# Risk of COVID-19 death in adults who received booster COVID-19 vaccinations in England

Isobel L. Ward[1] ✉, Chris Robertson [2,3], Utkarsh Agrawal[4], Lynsey Patterson[5,6], Declan T. Bradley [5,6], Ting Shi [7], Simon de Lusignan [4], F. D. Richard Hobbs[4,8], Aziz Sheikh [7] & Vahé Nafilyan [1]

The emergence of the COVID-19 vaccination has been critical in changing the course of the COVID-19 pandemic. To ensure protection remains high in vulnerable groups booster vaccinations in the UK have been targeted based on age and clinical vulnerabilities. We undertook a national retrospective cohort study using data from the 2021 Census linked to electronic health records. We fitted cause-specific Cox models to examine the association between health conditions and the risk of COVID-19 death and all-other-cause death for adults aged 50-100-years in England vaccinated with a booster in autumn 2022. Here we show, having learning disabilities or Down Syndrome (hazard ratio=5.07;95% confidence interval=3.69-6.98), pulmonary hypertension or fibrosis (2.88;2.43-3.40), motor neuron disease, multiple sclerosis, myasthenia or Huntington's disease (2.94, 1.82-4.74), cancer of blood and bone marrow (3.11;2.72-3.56), Parkinson's disease (2.74;2.34-3.20), lung or oral cancer (2.57;2.04 to 3.24), dementia (2.64;2.46 to 2.83) or liver cirrhosis (2.65;1.95 to 3.59) was associated with an increased risk of COVID-19 death. Individuals with cancer of the blood or bone marrow, chronic kidney disease, cystic fibrosis, pulmonary hypotension or fibrosis, or rheumatoid arthritis or systemic lupus erythematosus had a significantly higher risk of COVID-19 death relative to other causes of death compared with individuals who did not have diagnoses. Policy makers should continue to priorities vulnerable groups for subsequent COVID-19 booster doses to minimise the risk of COVID-19 death.

The development and rollout of COVID-19 vaccinations has significantly reduced severe COVID-19 infection, morbidity, and mortality[1]. However, some groups of patients remained at elevated risk of COVID-19 death despite having received a primary vaccination course. In addition, the effectiveness has been found to decline over time[2]. As a result, booster doses were needed to ensure that the population remains protected against severe outcomes, especially as restrictions controlling infections have been removed. In the UK, the first booster doses were offered to every adult three months after having a second COVID-19 vaccine dose[3]. Subsequent booster doses were then offered to groups at increased risk of hospitalisation or death, such as older adults, people with some health conditions, or

[1]Office for National Statistics, Newport, UK. [2]Department of Mathematics and Statistics, Strathclyde University, Glasgow, Scotland. [3]Public Health Scotland, Glasgow, Scotland. [4]Nuffield Department of Primary Care Health Sciences, University of Oxford, Oxford, UK. [5]Centre for Public Health, Queen's University Belfast, Belfast, UK. [6]Public Health Agency, Belfast, UK. [7]Usher Institute, University of Edinburgh, Edinburgh, UK. [8]NIHR Applied Research Collaboration, Oxford Thames Valley, Oxford, UK. ✉e-mail: Isobel.Ward@ons.gov.uk

individuals who were in contact with vulnerable patients, such as healthcare workers[4].

Continuing to protect the most vulnerable people against COVID-19 remains critical to minimise the direct and indirect impact of SARS-CoV-2 on population health and health services, especially as most countries transition to long-term management of the COVID-19 pandemic[5]. Identifying patients at greatest risk of severe outcomes from SARS-CoV-2 infection is important to ensure that the booster doses are offered to those who need them whilst managing the cost of the vaccination programme. Existing evidence suggests that besides age, having comorbidities and specific conditions such as chronic kidney disease (CKD) is associated with an increased risk of COVID-19 hospitalisation and death in individuals who had received a full vaccination schedule and a first booster dose[6]. However, there is little evidence on the risk factors for COVID-19 death for individuals who have received a second booster vaccination in autumn 2022 in England. Understanding the risk for different groups of individuals is imperative to inform the Joint Committee on Vaccination and Immunisation (JCVI) in the UK and equivalent international bodies decision making on prioritisation of subsequent booster vaccinations.

In this study, we aimed to identify the groups of adults who were at elevated risk of COVID-19 death, among those who had received a second booster dose of a COVID-19 vaccine as part of England's 2022 autumn booster campaign. Using a linked dataset based on the 2021 Census[7] linked to primary care records and death registration data, we estimated the hazard ratio of death involving COVID-19 for a range of sociodemographic characteristics and clinical risk factors. We also estimated the hazard ratio of death not involving COVID-19, to understand which groups were at higher risk of COVID-19 death but had otherwise a relatively low risk of death.

## Results

There were 14,651,440 adults aged 50-100 years in our study population (mean = 67.9 years, standard deviation (SD) = 10.9); 46.9% were male and 90.4% were White British. Descriptive characteristics of the study population are reported in Supplementary Data 1. Between September 1, 2022, and April 11, 2023, there were 6,800 COVID-19 deaths (52.2% male), and 150,075 non-COVID-19 deaths, 1 (48.4% male). The mean age of those who died from COVID-19 was 84.0 years (SD) = 8.87) and for non-COVID-19 deaths was 82.3 years (SD = 9.85).

Age was an important predictor of both COVID-19 and non-COVID-19 death; risk was 46 times greater for an 80-year-old relative to a 50-old for COVID-19-related death (HR:46.4;95%CI:43.0–50.0), and 30 times greater for non-COVID-19 related death (HR:29.6; 95% CI:29.2–30.0) (Fig. 1, Supplementary Data 2). Women were at a lower risk of COVID-19 death (HR:0.63; 95%CI 0.6–0.66) and non-COVID-19 death (HR:0.76; 95%CI:0.75–0.77) relative to men (Model 1) (Supplementary Data 2). We found a significantly lower risk of COVID-19

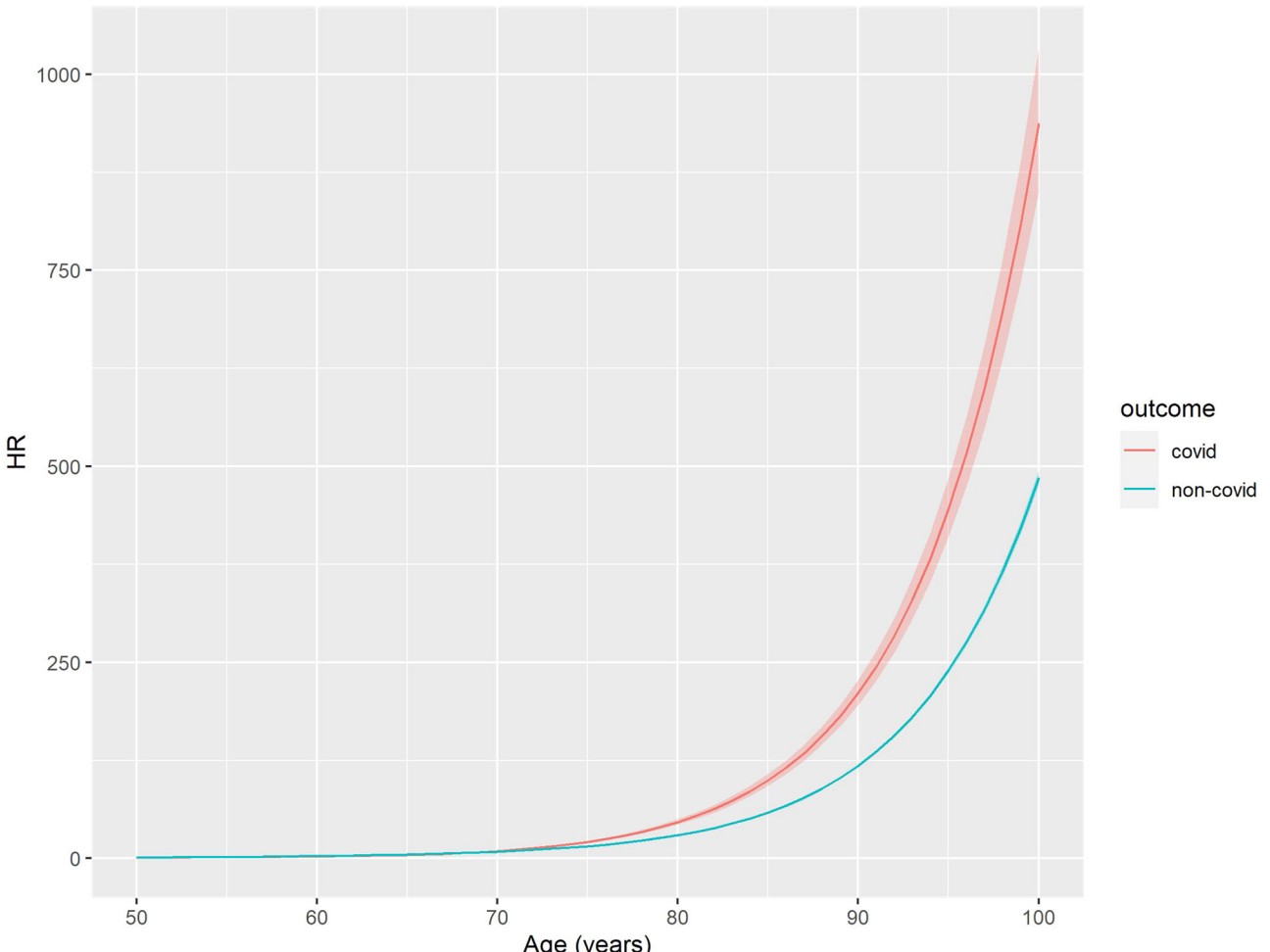

**Fig. 1 | Association between age and risk of COVID-19 death.** Hazard ratios (HRs) were calculated with a Cox regression model which was adjusted for age, sex and calendar time. The reference HR is for a 50-year-old male. COVID-19 outcomes are shown in red and non-COVID-19 outcomes in blue. The shaded area around the lines indicate the 95% confidence interval.

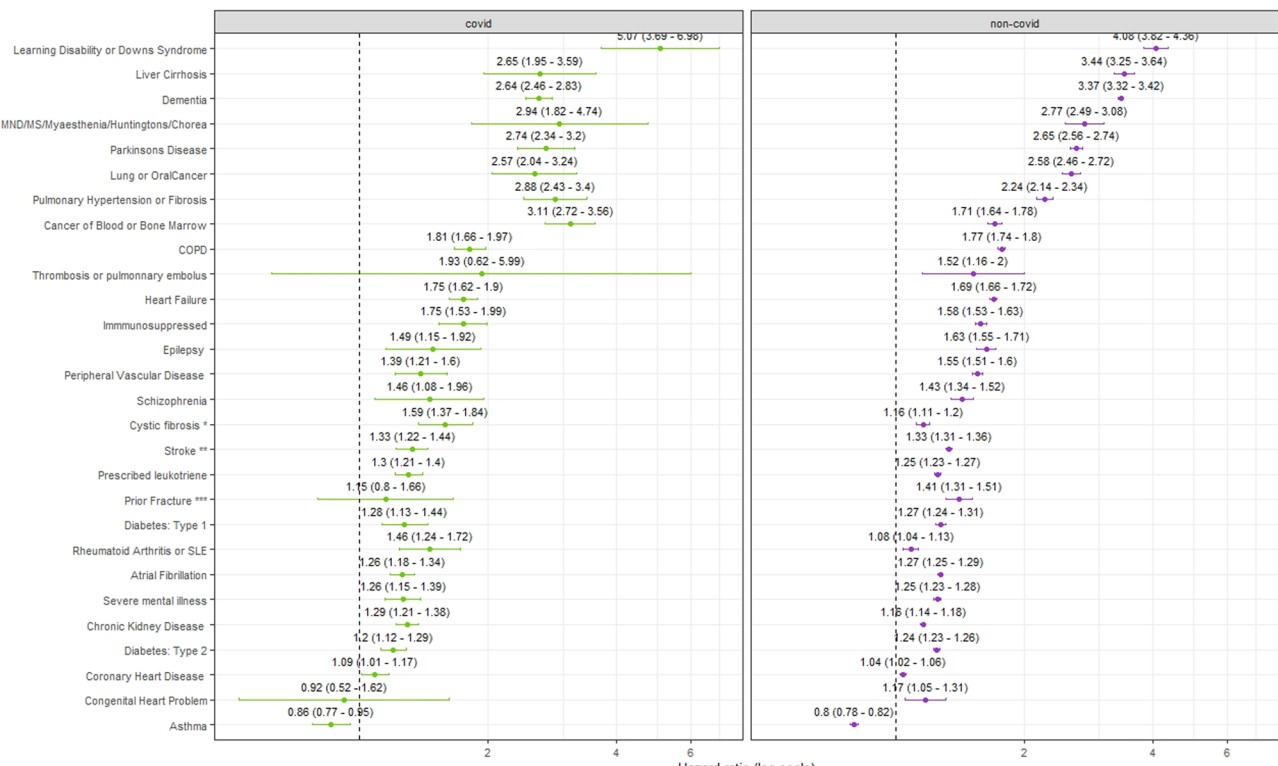

**Fig. 2 | Risk factors for COVID-19 and non-COVID-19 death in boosted adults.** Hazard ratios (HRs) were calculated with a Cox regression model which was adjusted for age, sex, calendar time, ethnicity, region and all other health conditions (Model 5). The dashed lines illustrate an HR of 1. The reference for all conditions is no diagnosis of the specified condition. The COVID-19 outcomes are shown in green and non-COVID-19 outcomes in purple. The HR is indicated with the point and the bars show the 95% confidence interval.

deaths for Black Caribbean (HR:0.46; 95%CI:0.29–0.73), Indian (HR:0.75;95%CI:0.58–0.95) and the other ethnic group (HR:0.62;95% CI:0.45–0.87) relative to the White British group for COVID-19 outcomes (Model 2) (Supplementary Data 2). Being morbidly obese or underweight is associated with an increased risk of COVID-19 (Morbidly obese = HR:1.60;95%CI:1.35–1.89, Underweight = HR:2.29;95% CI:2.04–2.59) and non-COVID-19 death (Morbidly obese = HR:1.55;95% CI:1.50–1.60, Underweight = HR:2.45;95%CI:2.39–2.51) (Model 3) (Supplementary Data 2). Self-reporting a disability which reduced an individual's day-to-day activities a little or a lot was associated with higher risk or COVID-19 and non-COVID-19 deaths relative to non-disabled people (Model 4) (Supplementary Data 2).

Overall, the conditions with the highest risk of COVID-death relative to not having the condition (HR > 2) were people with learning disabilities or Down's syndrome (HR:5.07;95%CI:3.69–6.98), those with pulmonary hypertension or fibrosis (HR:2.88;95%CI:2.43-3.40), motor neuron disease, multiple sclerosis, myasthenia or Huntington's disease (HR:2.94;95%CI:1.82–4.74), cancer of blood and bone marrow (HR:3.11;95%CI:2.72-3.56), Parkinson's disease (HR:2.74;95% CI:2.34–3.20), lung or oral cancer (HR:2.57;95%CI:2.04–3.24), dementia (HR:2.64;95%CI:2.46–2.83), or liver cirrhosis (HR:2.65;95%CI:1.95–3.59) (Fig. 2, Supplementary Data 2).

Comparing the HRs of COVID-19 and non-COVID-19 deaths, we found a higher relative risk of COVID-19 death compared to non-COVID-19 death for the following conditions: cancer of blood or bone marrow, CKD, cystic fibrosis, pulmonary hypotension or fibrosis or rheumatoid arthritis or systemic lupus erythematosus (SLE) (Supplementary table 2). Conversely, we found relative risk of dying from non-COVID-19 causes was higher than COVID-19 deaths for dementia and liver cirrhosis relative to those who do not have these conditions.

In the supplementary model not adjusted for other health comorbidities, the risk of COVID-19 outcomes was elevated relative to the model where the other conditions were adjusted for (Supplementary Table 3). Interestingly, for asthma in the model adjusted for other health conditions (Model 5) the HR for COVID-19 outcomes was lower than 1 (HR:0.84;95%CI:0.77-0.95), however in the supplementary model (adjusted for sex, age, region, ethnic group, and calendar time) the risk was significantly higher for people diagnosed with asthma compared to without (HR:1.22;95%CI:1.11-1.33) (Supplementary Table 3).

## Discussion

This national investigation has identified adults who remained at increased risk of COVID-19 death after receiving a second dose booster vaccination in England in Autumn 2022. Our results indicate that having learning disabilities or Down's syndrome, pulmonary hypertension or fibrosis, motor neuron disease, multiple sclerosis, myasthenia or Huntington's disease, cancer of blood and bone marrow, Parkinson's disease, lung or oral cancer, dementia or liver cirrhosis were independently associated with a higher risk of COVID-19 related death. For cancer of blood or bone marrow, CKD, cystic fibrosis, pulmonary hypotension or fibrosis or rheumatoid arthritis or SLE the increase in the relative risk was greater for COVID-19 death than non-COVID-19 death. Our findings suggest individuals in those groups were particularly vulnerable to COVID-19 death relative to other causes of death. For instance, for people with rheumatoid arthritis or SLE, the risk of dying from non-COVID-19 causes was not significantly different from people without these diagnoses; however, the risk was higher for COVID-19 death. Importantly, this group was not identified as one of the listed health comorbidities with the highest overall risk of COVID-19 death, but our analysis highlights the importance of relative risk

with individuals being more likely to die from COVID-19 in this group relative to other causes.

For many health conditions the increase in risk of COVID-19 death was similar to, or lower than, the increase in risk of non-COVID-19 deaths, suggesting that the increase in the risk of COVID-19 death was not different to the increase in the risk of death from other causes. Whilst we find that patients with asthma were at elevated risk of COVID-19 death after accounting for age, sex, ethnic group and region, we found that having asthma was not associated with the risk of COVID-19 death after adjusting for other comorbidities, suggesting that asthma was not directly increasing the risk of COVID-19 death.

Our findings support previous research which has assessed mortality outcomes following first dose COVID-19 booster vaccinations[8]. Overall, in the UK first dose booster vaccinations have been found to reduce severe outcomes (hospitalisation and death), with particular groups remaining at elevated risk[2]. Older adults (over 80 years of age), those with health comorbidities and specific conditions such as CKD were found to be at elevated risk. A study conducted in the United States reported that in patients who were immunocompromised, diabetic, had CKD or chronic lung disease there was a graded increase in risk of breakthrough COVID-19 infections positively associated with the number of comorbidities following two primary doses[9]. It is important to consider the results presented in our study may not reflect the differences in risk of COVID-19 death following infection. Our study looks at the risk of death since the time of having received a second booster dose, not since infection. It is possible that the risk of infection also differs by clinical risk factors, as patients who are the most vulnerable may maintain social isolation to protect themselves. It is also possible that some vulnerable patients may be at greater risk of infection because they live in communal establishments or have frequent contacts with carers or medical staff.

Critically, our work assesses the impact of the autumn 2022 booster vaccination on COVID-19, but additionally non-COVID-19 outcomes in adults in England. Our results provide strong evidence to inform JCVI about which groups should be prioritised for subsequent boosters and possibly subsequent boosters. It is critical to highlight the fact that some groups who do not have the overall highest risk of COVID-19 death, have an increased risk relative to non-COVID-19 causes and thus should remain a key priority.

Our study has several strengths. Firstly, we used population level data for England based on a unique linkage of the 2021 Census to electronic health records. Sociodemographic characteristics, including ethnic group, were derived from the 2021 Census, and were accurate and had low missingness, unlike in some electronic health records, where ethnic group is often missing and not always self-reported[10]. Second, we identified the clinical risk factors using primary care data. Third, we used information on the cause of death to define COVID-19 death and were also able to examine non-COVID-19 death as a comparator and identify which conditions were associated with a relative increased risk of larger for COVID-19 death than for non-COVID-19 death.

An important limitation of our study is the use of 2021 Census for our population means that people who did not respond to the Census were excluded. In addition, it also excluded Census respondents who could not be linked to the Personal Demographics Service (PDS). However, the data we used covered 96.0% of those who received a booster dose in England the autumn of 2022. One of the limitations of our work was the lack of data on COVID-19 hospital admissions[11]. In order to effectively manage resource and understand which groups are at the highest risk of hospitalisation, subsequent work with access to timely data should account for hospital admittance. Additionally, we are unable to account for behaviours which would be classified as health protective such as minimising social contact in the present study. Therefore, it is important to consider for some patients whose risk of hospitalisation or death was most pertinent following SARS-

CoV-2 infection, they may be maintaining social isolation to protect themselves. Hence for groups of individuals where the risk was not higher for COVID-19 outcomes, but overall was greater for all-cause death we must maintain prioritisation of vaccination to these individuals. Subsequent research should explore common conditions (e.g., asthma) to understand if the interaction between having a common diagnosis in addition to another specific condition results in a particular susceptibility to adverse COVID-19 outcomes.

Our work investigates the risk of cause-specific COVID-19 death, as well as non-COVID-19 death in a cohort of adults who received a booster dose in the autumn of 2022. In order to effectively manage the COVID-19 risk, it is imperative that the most vulnerable groups of individuals are prioritised for COVID-19 booster vaccinations. We highlight that the risk of COVID-19 death, compared to all other cause death, remains particularly high in adults with learning disabilities or Down syndrome, pulmonary hypertension or fibrosis, motor neuron disease, multiple sclerosis, myasthenia or Huntington's disease, cancer of blood and bone marrow, Parkinson's disease, lung or oral cancer, dementia, or liver cirrhosis. These groups of patients should be a key priority for subsequent vaccinations, therapeutics, and novel treatment. In addition, we highlight the risk associated with a range of health conditions and sociodemographic characteristics which should inform policy makers and researchers with key demographics of interest for subsequent research and vaccination.

## Methods
This study was approved by the National Statistician's Data Ethics Advisory Committee (NSDEC) (UK Statistics Authority).

### Data sources
We used person-level data comprising individuals in the 2021 Census, linked to the Personal Demographics Service (PDS)[12], to obtain NHS numbers with a linkage rate of 94.6%. These individuals were then linked via NHS number to Office for National Statistics(ONS) death registrations[13] (including deaths registered up to April 26, 2023) and other electronic health records. Vaccination records were obtained from the National Immunisation Management Service (NIMS)[14]. In the UK in autumn 2022 all vaccinations administered to adults (18 years and older) were mRNA doses given by GPs, vaccination centres and community pharmacists. Health related variables from primary care records were derived from the General Practice Extraction Service (GPES) data for Pandemic Planning and Research version 4 (GDPPR) and linked via NHS number. The linked dataset included data on 52 million people residents in England, which covers (accounting for linkage and non-respondents) approximately 91.8% of the population of England on Census day in 2021.

### Study population
This study was a retrospective cohort study that included individuals vaccinated with a COVID-19 booster dose in England after September 1, 2022, were enumerated in Census 2021, linked to the PDS, and were aged 50-100-years of age on the date of booster administration. An autumn booster dose was defined as a booster dose administered on or after September 1, 2022, and at least 84 days since the last clinically acceptable dose, and individuals must have had at least two doses prior to the booster (Supplementary Figure 1). See supplementary Table 1 for sample flow.

### Outcome
The primary outcome of this study was COVID-19 death, defined as any International Classification of Diseases (ICD) death with codes U07.1 and U07.2[15] recorded anywhere on the death certificate, which occurred up to April 11, 2023, and were registered by April 26, 2023 (allowing for 15 days for death registration). The secondary outcome was all-cause non-COVID-19 deaths.

Individuals were followed from 14-days after the autumn dose booster vaccination date until April 11, 2023. Time at risk started 14 days after booster vaccination and ended at time of death (either COVID-19 or other), or end of study (April 11, 2023).

## Exposures

The predictors included in the model included sociodemographic characteristics from Census 2021 and clinical risk factors from primary care data. The medical conditions were derived using primary care records from General Practice Extraction Service (GPES) data for Pandemic Planning and Research version 4 (GDPPR) based on the definitions used by the QCovid2[16,17], risk prediction model, using data between March 1, 2015, and March 21, 2021 (Census day). The QCovid risk prediction model was used by the NHS in Great Britain to identify individuals who are particularly vulnerable to COVID-19 hospitalisation and death. The model has been validated previously using the PHDA[17] and data from Scotland[18]. All predictors included in the model are summarised in Supplementary Data 1. Missing items for Census characteristics were imputed using nearest-neighbour donor imputation, the methodology employed by ONS across all 2011 Census variables. Missing Body Mass Index (BMI) information was handled by including a 'Missing' category.

## Statistical analysis

We used cause-specific Cox regression to examine the association between each of the health comorbidities and the risk of COVID-19 death and all-other-cause mortality. Time at risk started 14 days after receipt of the autumn booster dose and ended either at time of death or end of study period (April 11, 2023). For each outcome (COVID-19 and non-COVID-19), we first fitted a model adjusted for age (restricted cubic spline with boundary knots at the $5^{th}$ and $95^{th}$ percentile with 3 internal knots), sex and calendar time (restricted cubic spline with boundary knots at the $5^{th}$ and $95^{th}$ percentile with 3 internal knots) (Model 1), next added ethnic group and region (Model 2), BMI (Model 3) and disability (Model 4).

For each of the health comorbidities we fitted a model adjusted for all QCovid health conditions, age (spline), sex, calendar time (spline), ethnic group and region (Model 5). In addition, we fitted separate models for each health conditions, adjusted for age (restricted cubic spline), sex, calendar time (restricted cubic spline), ethnic group and region. For computational reasons, the analysis was performed on a dataset containing every individual who died and a weighted 5% sample of those who survived up until the end of study. To test for differences in risk between COVID-19 and non-COVID-19 models for people who had diagnoses of health comorbidities, Z-statistics were calculated for each coefficient from models (Model 5) with bootstrapped data (1000 iterations). The bootstrapping was run on the sampled data ensuring the proportion of outcomes (death and alive) was preserved. Analysis was conducted using Spark version 2.4.0 and R version 3.5.1. All counts have been rounded to the nearest 5, with values less than 10 suppressed for disclosure reasons.

## Reporting summary

Further information on research design is available in the Nature Portfolio Reporting Summary linked to this article.

## Data availability

The source data used in this study is subject to controlled access due to its sensitive nature. All statistical data used in this study are available from the Office for National Statistics website. Source data are provided with this paper.

## Code availability

Code used in this study is available on Github [link]

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

## Acknowledgements

The authors would like to thank Daniel Ayoubkhani for their statistical expertise and methodological guidance. This work has been funded by UKRI as part of the award to the University of Birmingham for COVID-19 National Core Studies (NCS), for "Phase 1 COVID-19 Immunity – National Core Study (Phase 1 IMM-NCS)". The award reference number is MC_PC_20061.

## Author contributions

I.W., V.N. and A.S. conceptualised and designed the study. I.W. prepared the study and performed the statistical analysis, which were quality checked by V.N., I.W., V.N., C.R., and A.S., contributed to interpretation of the findings. I.W. and V.N. wrote the original draft. UA, LP, DB, TS, SdeL and RH contributed to reviewing and editing of the manuscript and approved the final manuscript.

## Competing interests

SdeL has received University funding for vaccine related research from AstraZeneca, GSK, Moderna, Pfizer, Sanofi, Seqirus, and Takeda, and been advisory board members for AstraZeneca, GSK, Sanofi and Seqirus. The remaining authors declare no competing interests.
