## [Peer Review File · Nature Communications]

REVIEWER COMMENTS

Reviewer #1 (Remarks to the Author):

Thank you for the opportunity to review this timely analysis. This is an important and useful study of the risk of COVID-19 death in a large national cohort. The study is well-conducted and the findings are of global interest and inform decisions about booster vaccination. I have suggested few revisions in the spirit of providing further clarifications of the implications of these findings.

1) How did the COVID-19 mortality findings at this stage of the pandemic differ from earlier stages in this same cohort? It would be useful to provide some context and information to appreciate these results. Did you observe also a major transition in COVID-19 fatality after Omicron (see for example Chemaitelly, H., et al. A turning point in COVID-19 severity and fatality during the pandemic: A national cohort study in Qatar. medRxiv, 2023.2005.2028.23290641 (2023)).

2) It would be useful to try to link these findings and interpret them in terms of the drivers of COVID-19 mortality: viral virulence deaths, forward displacement of mortality, and/or Long COVID. See for example Chemaitelly, H., et al. Short- and longer-term all-cause mortality among SARS-CoV-2- infected persons and the pull-forward phenomenon in Qatar. medRxiv, 2023.2001.2029.23285152 (2023). Such implications would be useful to understand the actual drivers of mortality at this stage of the pandemic and thus inform programmatic and policy decisions.

3) I do not seem to find the details of the used boosters. It would be useful to provide further information on the type of boosters used in autumn 2022 and how booster vaccination was conducted. It might be also useful to link to recent findings about the effectiveness of these boosters.

4) It seems there were last minute changes to the text, but these changes were not sufficiently polished, such as existence of missing numbers “risk was xx times 203 greater for an 80-year-old relative to a 50-old for COVID-19 related death (HR:46.4;95%CI:43.0-50.0), 204 and xx times greater for non-COVID-19 related death (HR:29.6; 95%CI:29.2-30.0) (Figure 1, Table 2).” It would be best to carefully proofread the manuscript and fix all such polishing issues.

5) In some of the figures, particularly Figure 1, the colors are so similar that makes it difficult to understand the legend. Please distinguish the curves to avoid confusion.

Reviewer #2 (Remarks to the Author):

The aim of the manuscript Risk of COVID-19 death in adults who received booster COVID-19 vaccinations: national retrospective cohort study on 14.6 million people in England is to identify adults who had received a booster vaccination as part of the autumn 2022 campaign in England yet remained at increased risk of postbooster COVID-19 death and compared to non-COVID-19 risk. This descriptive

study has some potential implications for priorities in Covid-19 vaccinations. However, I do not believe that the study is innovative enough to be published in Nature Communications. The focus on a group that received a booster Covid-19 vaccination is interesting but does not add enough to the study of risk factors/risk groups of Covid-19 mortality when compared to many other studies in the field.

- The primary aim of the study seems somewhat unclear. I was expecting a study comparing the risk of Covid-19 death among risk groups who received a booster vaccination and those who did not. However, the study focuses only on those who received a booster vaccination. In my view the study therefore lacks an important dimension especially if it claims that results have important implications for prioritisation of vaccination booster doses worldwide.
- Another severe outcome would add to the robustness of findings. Did the authors have access to data on intensive care/hospitalization due to Covid-19 complications?
- The authors estimated the hazard ratio of death not involving COVID-19, to understand which groups were at higher risk of COVID-19 death but had otherwise a relatively low risk of death. What is the risk of misclassification of deaths?
- To me it is unclear why the authors decided to focus at certain risk groups/health conditions in favor of others. The various diseases/health conditions studied have very different etiology which makes it difficult to make any interpretations of potential causal pathways.
- Some numbers missing starting on line 202? "Age was an important predictor of both COVID-19 and non-COVID-19 death; risk was xx times greater..."
- The authors found a significantly lower risk of COVID-19 deaths for ethnic minority groups relative to the White British group for COVID-19 outcomes. This is interesting and seems to contradict some UK studies on ethnicity and Covid-19 that found higher Covid-19 mortality by ethnicity although associations vary by specific ethnic group and wave. Does this rather solid finding suggest that a booster Covid-19 is more protective in ethnic minorities?
- The authors argue that their results provide strong evidence to inform JCVI about which groups should be prioritised for the autumn 2023 and possibly subsequent boosters. This statement seems somewhat dated since it is already autumn of 2023.

Response to reviewers: Risk of COVID-19 death in adults who received booster COVID-19 vaccinations: national retrospective cohort study on 14.6 million people in England

We would like to thank the reviewers for their helpful responses and for taking the time to review our publication. Please see our responses to each of the comments raised below. We have amended our publication using tracked changes.

Reviewer #1 (Remarks to the Author):

Thank you for the opportunity to review this timely analysis. This is an important and useful study of the risk of COVID-19 death in a large national cohort. The study is well-conducted and the findings are of global interest and inform decisions about booster vaccination. I have suggested few revisions in the spirit of providing further clarifications of the implications of these findings.

- 1) How did the COVID-19 mortality findings at this stage of the pandemic differ from earlier stages in this same cohort? It would be useful to provide some context and information to appreciate these results. Did you observe also a major transition in COVID-19 fatality after Omicron (see for example Chemaitelly, H., et al. A turning point in COVID-19 severity and fatality during the pandemic: A national cohort study in Qatar. medRxiv, 2023.2005.2028.23290641 (2023)).
- 2) It would be useful to try to link these findings and interpret them in terms of the drivers of COVID-19 mortality: viral virulence deaths, forward displacement of mortality, and/or Long COVID. See for example Chemaitelly, H., et al. Short- and longer-term all-cause mortality among SARS-CoV-2- infected persons and the pull-forward phenomenon in Qatar. medRxiv, 2023.2001.2029.23285152 (2023). Such implications would be useful to understand the actual drivers of mortality at this stage of the pandemic and thus inform programmatic and policy decisions.

We thank the reviewer for flagging these two pre-print articles. We note the importance of comparing our results in the context of worldwide literature. We have discussed our results in the context of other literature evaluating the impact of vaccinations on the COVID-19 pandemic. The following is in the discussion:

Our findings support previous research which has assessed mortality outcomes following first dose COVID-19 booster vaccinations [16]. Overall, in the UK first dose booster vaccinations have been found to reduce severe outcomes (hospitalisation and death), with particular groups remaining at elevated risk [2]. Older adults (over 80 years of age), those with health comorbidities and specific conditions such as CKD were found to be at elevated risk. A study conducted in the United States reported that in patients who were immunocompromised, diabetic, had CKD or chronic lung disease there was a graded increase in risk of breakthrough COVID-19 infections

positively associated with the number of comorbidities following two primary doses [17].

Additional research is required to understand the impact Long COVID has on mortality outcomes.

3) I do not seem to find the details of the used boosters. It would be useful to provide further information on the type of boosters used in autumn 2022 and how booster vaccination was conducted. It might be also useful to link to recent findings about the effectiveness of these boosters.

This is an important point and we have now added some additional information to the Methods section detailing the boosters administered in England for the autumn 2022 campaign.

*Vaccination records were obtained from the National Immunisation Management Service (NIMS) [10]. **In the UK in autumn 2022 all vaccinations administered to adults (18 years and older) were mRNA doses given by GPs, vaccination centres and community pharmacists.** Health related variables from primary care records were derived from the General Practice Extraction Service (GPES) data for Pandemic Planning and Research version 4 (GDPPR) and linked via NHS number.*

4) It seems there were last minute changes to the text, but these changes were not sufficiently polished, such as existence of missing numbers “risk was xx times 203 greater for an 80-year-old relative to a 50-old for COVID-19 related death (HR:46.4;95%CI:43.0-50.0), 204 and xx times greater for non-COVID-19 related death (HR:29.6; 95%CI:29.2-30.0) (Figure 1, Table 2).” It would be best to carefully proofread the manuscript and fix all such polishing issues.

We apologise for this oversight in the text of our manuscript. The inline text values have now been updated.

*Age was an important predictor of both COVID-19 and non-COVID-19 death; risk was **46** times greater for an 80-year-old relative to a 50-old for COVID-19 related death (HR:46.4;95%CI:43.0-50.0), and **30** times greater for non-COVID-19 related death (HR:29.6; 95%CI:29.2-30.0) (Figure 1, Table 2).*

5) In some of the figures, particularly Figure 1, the colours are so similar that makes it difficult to understand the legend. Please distinguish the curves to avoid confusion.

In Figure 1, we have shown the hazard ratio for COVID-19 outcomes in red and non-COVID-19 outcomes in blue. The reference is a 50-year-old male. There is very little risk in outcomes between COVID-19 and non-COVID-19 deaths in adults under 70 years of age. The hazard ratios for each decile of year for males and females are reported in Table 2.

Reviewer #2 (Remarks to the Author):

The aim of the manuscript Risk of COVID-19 death in adults who received booster COVID-19 vaccinations: national retrospective cohort study on 14.6 million people in England is to identify adults who had received a booster vaccination as part of the autumn 2022 campaign in England yet remained at increased risk of postbooster COVID-19 death and compared to non-COVID-19 risk. This descriptive study has some potential implications for priorities in Covid-19 vaccinations. However, I do not believe that the study is innovative enough to be published in Nature Communications. The focus on a group that received a booster Covid-19 vaccination is interesting but does not add enough to the study of risk factors/risk groups of Covid-19 mortality when compared to many other studies in the field.

- The primary aim of the study seems somewhat unclear. I was expecting a study comparing the risk of Covid-19 death among risks groups who received a booster vaccination and those who did not. However, the study focuses only on those who received a booster vaccination. In my view the study therefore lacks an important dimension especially if it claims that results have important implications for prioritisation of vaccination booster doses worldwide.

The primary goal of this study was to assess which groups remain most vulnerable to COVID-19 mortality following booster vaccination. This study was conducted to inform the rollout of further doses, which should be preferentially targeted at the groups that remain at elevated risk of COVID-19 death despite having received a booster. Whilst we agree that a study comparing the risk of death between people who had received a booster and those who had not would be interesting, it would be a different study with a different aim (eg measuring the effectiveness of the booster, rather than identifying groups that remained at elevated risk despite being boosted).

•Another severe outcome would add to the robustness of findings. Did the authors have access to data on intensive care/hospitalization due to Covid-19 complications?

For this analysis, we did not have access to hospitalisation data. We have highlighted the section where we have outlined this in our limitations section.

One of the limitations of our work was the lack of data on COVID-19 hospital admissions [19]. In order to effectively manage resource and understand which groups are at the highest risk of hospitalisation, subsequent work with access to timely data should account for hospital admittance.

•The authors estimated the hazard ratio of death not involving COVID-19, to understand which groups were at higher risk of COVID-19 death but had otherwise a relatively low risk of death. What is the risk of misclassification of deaths?

The risk of misclassification of death is low in England and Wales as deaths are certified by a doctor [1*]. Mortality statistics are based on information recorded when deaths are certified and registered. All of the conditions mentioned on the death certificate are coded using the International Classification of Diseases, Tenth Revision (ICD-10) is used. From all causes listed an underlying cause of death is selected using ICD-10 coding rules. The underlying cause is defined by the World Health Organisation as the disease or injury that initiated the train of events directly leading to death or the circumstances of the accident/violence that produced the fatal injury. The statistics we use for COVID-19 deaths rely solely on death certification. We believe this is a strength of our study in comparison to other COVID-19 studies which use death within 28 days of a positive COVID-19 test as the definition for inclusion.

[1*]

<https://www.ons.gov.uk/peoplepopulationandcommunity/birthsdeathsandmarriages/deaths/methodologies/userguidetomortalitystatisticsjuly2017#:~:text=Deaths%20should%20be%20registered%20within,then%20investigated%20by%2C%20a%20coroner>

•To me it is unclear why the authors decided to focus at certain risk groups/health conditions in favour of others. The various diseases/health conditions studied have very different etiology which makes it difficult to make any interpretations of potential causal pathways.

The certain groups were identified as risk factors in our study were selected based on the QCovid prediction model, which was developed at the request of the UKs Chief Medical Officers to inform deliberations on risk stratification [2*]. QCovid is a coronavirus risk prediction model created by the University of Oxford which was used to identify medical conditions with increased risk related to COVID-19. A description of the QCovid risk prediction model has been included in the methods section. I have included the text below.

The QCovid risk prediction model was used by the NHS in Great Britain to identify individuals who are particularly vulnerable to COVID-19 hospitalisation and death.

The model has been validated previously using the PHDA [14] and data from Scotland [15].

[2*] <https://qcovid.org/>

•Some numbers missing starting on line 202? “Age was an important predictor of both COVID-19 and non-COVID-19 death; risk was xx times greater...

We apologise for this oversight in the text of our manuscript. The inline text values have now been updated.

*Age was an important predictor of both COVID-19 and non-COVID-19 death; risk was **46** times greater for an 80-year-old relative to a 50-old for COVID-19 related death (HR:46.4;95%CI:43.0-50.0), and **30** times greater for non-COVID-19 related death (HR:29.6; 95%CI:29.2-30.0) (Figure 1, Table 2).*

•The authors found a significantly lower risk of COVID-19 deaths for ethnic minority groups relative to the White British group for COVID-19 outcomes. This is interesting and seems to contradict some UK studies on ethnicity and Covid-19 that found higher Covid-19 mortality by ethnicity although associations vary by specific ethnic group and wave. Does this rather solid finding suggest that a booster Covid-19 is more protective in ethnic minorities?

Looking at the results for ethnic group the Black Caribbean (HR: 0.46, 95%CI: 0.29-0.73) and Indian (HR: 0.75, 95% CI: 0.58-0.96) were the only groups to have a significantly lower risk for COVID-19 death than the White British group. For the other ethnic groups, the confidence estimate indicates that the lower risk indicated from the estimate is not significant. We have not focused on this finding in our discussion given the difference in the underlying populations of the vaccinated cohorts by ethnic group. Uptake of the vaccination and subsequent boosters is known to vary by ethnic group and to formally test your proposal of a booster COVID-19 dose being more protective in ethnic minorities we would need to account for additional confounding in our study. This is however an important point which should be the focus of subsequent research.

•The authors argue that their results provide strong evidence to inform JCVI about which groups should be prioritised for the autumn 2023 and possibly subsequent boosters. This statement seems somewhat dates since it is already autumn of 2023.

Thank you for this comment. We have now revised our text to highlight that this is appropriate for subsequent booster prioritisation. This comment was flagged as at the time of analysis this unpublished work was presented to JCVI to inform the roll out of the autumn 2023.

Critically, our work is the first to assess the impact of the autumn 2022 booster vaccination on COVID-19, but additionally non-COVID-19 outcomes in adults in England. Our results provide strong evidence to inform JCVI about which groups should be prioritised for subsequent boosters and possibly subsequent boosters. It is critical to highlight the fact that some groups who do not have the overall highest risk

of COVID-19 death, have an increased risk relative to non-COVID-19 causes and thus should remain a key priority.